# Corticosterone Impairs Hippocampal Neurogenesis and Behaviors through *p21*-Mediated ROS Accumulation

**DOI:** 10.3390/biom14030268

**Published:** 2024-02-23

**Authors:** Guanhao Wang, Lining Cao, Shuanqing Li, Meihui Zhang, Yingqi Li, Jinjin Duan, You Li, Zhangsen Hu, Jiaan Wu, Tianming Li, Ming Jiang, Jianfeng Lu

**Affiliations:** 1Shanghai YangZhi Rehabilitation Hospital, Shanghai Sunshine Rehabilitation Center, Frontier Science Center for Stem Cell Research, School of Life Sciences and Technology, Tongji University, Shanghai 200092, China; 2Institute of Biophysics, Chinese Academy of Sciences, Beijing 100045, China; 3College of Life Sciences, University of Chinese Academy of Sciences, Beijing 100045, China; 4Suzhou Institute, Tongji University, Suzhou 200070, China

**Keywords:** corticosterone, *p21*, neurogenesis, anxiety-like behaviors

## Abstract

Stress is known to induce a reduction in adult hippocampal neurogenesis (AHN) and anxiety-like behaviors. Glucocorticoids (GCs) are secreted in response to stress, and the hippocampus possesses the greatest levels of GC receptors, highlighting the potential of GCs in mediating stress-induced hippocampal alterations and behavior deficits. Herein, RNA-sequencing (RNA-seq) analysis of the hippocampus following corticosterone (CORT) exposure revealed the central regulatory role of the *p21 (Cdkna1a)* gene, which exhibited interactions with oxidative stress-related differentially expressed genes (DEGs), suggesting a potential link between *p21* and oxidative stress-related pathways. Remarkably, *p21*-overexpression in the hippocampal dentate gyrus partially recapitulated CORT-induced phenotypes, including reactive oxygen species (ROS) accumulation, diminished AHN, dendritic atrophy, and the onset of anxiety-like behaviors. Significantly, inhibiting ROS exhibited a partial rescue of anxiety-like behaviors and hippocampal alterations induced by *p21*-overexpression, as well as those induced by CORT, underscoring the therapeutic potential of targeting ROS or *p21* in the hippocampus as a promising avenue for mitigating anxiety disorders provoked by chronic stress.

## 1. Introduction

Anxiety disorders refer to a group of costly psychiatric disorders characterized by feelings of fear, panic, and worry and related behavioral disturbances. Up to 275 million people worldwide (4% of the global population) are affected by anxiety disorders, which were ranked 8th among the top 25 leading causes of years lived with disability (YLDs) in 2019 [1]. However, around 50% of patients diagnosed with generalized anxiety disorder (GAD) do not respond to conventional pharmacological and psychological interventions. Moreover, the limited accessibility of mental health professionals, the risk of dependence on benzodiazepines, and the side effects of selective serotonin reuptake inhibitors (SSRIs) can negatively influence medication adherence in patients. Hence, there is an urgent need to investigate the underlying pathological mechanisms and develop innovative therapeutic interventions for GAD.

People who have lived with chronic stress events, such as a stressful environment, chronic illness, or child abuse, are more likely to develop GAD. In animal studies, chronic stress is known to trigger histological alterations in the hippocampus, such as reduced adult hippocampal neurogenesis (AHN) [2,3] and altered dendritic complexity of the neurons in the hippocampal dentate gyrus (DG) area [4,5], though the underlying mechanism is complex and not fully understood. The secretion of glucocorticoids (GCs, corticosterone in rats, and cortisol in humans) is a primary endocrine response to stress. It had been reported that the mean cortisol level increased around nine times during stressful moments compared to relaxed periods [6]. The hippocampus is well known as the principal target of GCs in the brain since it contains the greatest concentration of GC receptors. This raises the possibility that stress-induced GC elevation might significantly influence the structures of hippocampal neurons and their functions. 

Indeed, as the end-effectors for the hypothalamic–pituitary–adrenal axis, GCs are well documented to exert effects overlapping with those induced by chronic stress exposure on hippocampal structure and function [7,8,9], leading to suggestions that GC secretion is critical in stress-induced hippocampal damage. Although the direct influence of GCs on the hippocampus [7,10] and the development of anxiety deficits [11,12] has been elucidated in numerous studies, the underlying mechanisms by which prolonged GC exposure induces hippocampal damage and the associated behavioral deficits remain to be elucidated. 

Animal studies have demonstrated that chronic stress induces histological alterations in the hippocampus [2,3,4,5] and anxiety-like behaviors, but the intricate mechanisms underlying these changes remain incompletely understood. Given that the endocrine response to stress is the secretion of GCs, with corticosterone (CORT) in rodents and cortisol in human beings, and the hippocampus is the principal target site in the brain for GCs, understanding the molecular events occurring in the hippocampus following prolonged exposure to CORT is crucial for unraveling the etiological mechanisms of GAD. This study aims to elucidate the link between prolonged CORT exposure and anxiety- and depressive-like behaviors, accompanied by histological alterations in the hippocampus. Through RNA-sequencing (RNA-seq) analysis of the CORT-treated hippocampus, we identified the hub gene *p21*, which exhibited interactions with oxidative stress-associated differentially expressed genes (DEGs). Overexpression of *p21* (*p21*-OE) in the hippocampal dentate gyrus was found to partially recapitulate CORT-induced phenotypes, including reactive oxygen species (ROS) accumulation, reduced adult hippocampal neurogenesis (AHN), dendritic atrophy, and the manifestation of anxiety-like behaviors. Notably, the inhibition of ROS partially attenuated anxiety-like behaviors and hippocampal alterations induced by *p21*-OE, as well as phenotypes induced by CORT, highlighting the potential of targeting ROS or *p21* in the hippocampus as a therapeutic approach to mitigate mood disorders induced by chronic stress.

## 2. Materials and Methods

### 2.1. Housing of Animals

All interventions and animal care procedures have been approved by the Animal Committee of the School of Life Sciences and Technology, Tongji University, Shanghai, China. Male C57BL6/J mice (age: 6–8 weeks; weight: 23–25 g) were purchased from Charles River Laboratories (Beijing, China), and housed in a Specific Pathogen Free (SPF) animal facility. They were subjected to standard conditions (room temperature: 24 °C; humidity: 55%) under a 12:12 light/dark cycle. Mice were given ad libitum access to food and water. 

### 2.2. Prolonged Exposure to Corticosterone in Mice

For drug trials, 34.7 mg of corticosterone (T0948L, TargetMol, Boston, MA, USA) was dissolved in 1 mL of DMSO, and then 18.8 mL of sterile water containing 200 μL of Tween-80 was added to the solution to reach a final concentration of 1.734 mg/mL. Adult male C57BL/6 mice (age 7–8 weeks) were randomly divided into 2 groups (vehicle: n = 32; CORT: n = 48), and they were administered with CORT (40 mg/kg) or vehicle (23.07 mL/kg) every day via subcutaneous (s.c.) injection for 36 days. 

### 2.3. Manipulation of p21 Expression in Hippocampal Dentate Gyrus by Stereotaxic Injection of AAV Vectors 

The recombinant AAV vectors (*p21* overexpression vector: pAAV-CMV-Cdkn1a-3xFLAG-P2A-mNeonGreen-tWPA; negative control for *p21* overexpression vector: pAAV-cmv-3xFLAG-P2A-mNeonGreen-tWPA; *p21* knock-down vector: pscAAV-U6-shRNA (Cdkn1a)-CMV-EGFP-tWPA; negative control for *p21* knock-down vector: pscAAV-U6-shRNA (NC2)-CMV-EGFP-tWPA) were purchased from OBiO Technology (Shanghai, China).

Adult male C57BL/6 mice (age 7–8 weeks) were randomly divided into 4 groups (*p21*-OE/NC, *p21*-OE, *p21*-shRNA, NC-shRNA, n = 16 per group), and then the AAV vectors were delivered into the hippocampal DG region of mice to manipulate *p21* expression. For stereotaxic injection, mice were anesthetized by using isoflurane inhalation anesthesia (3% during induction, 1.5% during maintenance, flow rate: 2 L/min). AAV vectors were injected into the target sites (bilateral DG: −1.9 mm AP, ±1.2 mm ML, −2.25 mm DV) by using a 10-μL Hamilton syringe (0.5 μL/5 min). The injection doses for AAV-based viral vectors ranged from 2.4 × 1012 to 3.6 × 1012 viral genomes (vg)/mL. The needle was kept in situ for an additional 10 min to prevent solution leakage. Then the needle was removed gently following injection. 

### 2.4. Antioxidant Treatment Study

The antioxidant N-acetylcysteine (NAC) (HY-B0215, MedChemExpress, Monmouth Junction, NJ, USA) was administered to mice via drinking water (1 g/L) as previously described [13].

In order to examine the therapeutic effect of antioxidant on *p21*-overexpression-induced hippocampal damage and behavioral deficits, adult male C57BL/6 mice (age 7–8 weeks) were randomly divided into 4 groups to receive *p21*-OE or negative control AAV vectors with or without antioxidant treatment (n = 16 per group). 

In order to examine the therapeutic effect of antioxidant on CORT-induced hippocampal damage and behavioral deficits, adult male C57BL/6 mice (age 7–8 weeks) were randomly divided into 4 groups. They were administered with vehicle or CORT with or without antioxidant treatment (n = 16 per group). 

### 2.5. Behavioral Tests

Mice were allowed to acclimate to the testing room for at least 30 min prior to the test. Experimenters were blinded to the treatment conditions of mice. In order to avoid the variables specific to each session of the experiment, the subject of each group will be represented in each session of the behavioral test. Between each test session, the apparatuses were cleaned with a 75% ethanol solution to minimize the influence of olfactory cues. For behavioral tests, the movements of animals were videotaped and analyzed using a SMART video-tracking system (Panlab Harvard apparatus, Barcelona, Spain). 

#### 2.5.1. Open Field Test (OFT)

The open field test was performed in the apparatus (40 cm × 40 cm × 30 cm) made of gray plastic. The mice were initially placed in the center zone, and they were allowed to freely explore the whole arena for 10 min. The total distance traveled, time spent in the central area, and entries into the central area were analyzed during the 10 min period of the test.

#### 2.5.2. Elevated plus Maze Test (EPMT)

The elevated plus maze apparatus comprised 2 open arms (30 cm × 5 cm × 0.5 cm), 2 closed arms (30 cm × 5 cm × 16 cm), and a central area. Each mouse was allowed to explore the apparatus freely for 10 min after being placed facing an open arm initially in the central zone. The time spent and distance traveled in the open arms were then analyzed.

#### 2.5.3. Tail Suspension Test (TST)

The TST was performed as previously described elsewhere [14]. The mouse was lifted gently by its tail and hanged on a horizontal bar 50 cm above the ground using adhesive tape, allowing it to hang freely without touching any surfaces. The duration of immobility was recorded during the 6 min test.

#### 2.5.4. Forced Swimming Test (FST)

Each mouse was placed in a transparent cylindrical water tank (30 cm height × 15 cm diameter) containing 15 cm of water at 25 °C for 6 min. The floating time, during which the mouse only kept a state of inactivity with its head at the water surface or made slight movements, was recorded to determine the duration of immobility during the last 4 min of the 6 min test. 

### 2.6. DHE Administration

The mice received an intraperitoneal\injection of DHE (27 mg/kg, 10 mg/mL, 40% DMSO in PBS) 18 h before anesthetization. Subsequently, the mice were transcardially perfused with saline, followed by 4% paraformaldehyde (PFA). Brain samples were post-fixed in 4% PFA at 4 °C overnight and then cryo-protected in 30% sucrose for an additional 24 h at 4 °C. The brains were embedded in O.C.T. compound prior to cryostat sectioning. The frozen tissue blocks were sectioned into slices with 20 μm thickness using a cryostat (CM1950, Leica, Wetzlar, Germany).

### 2.7. Transcardial Perfusion

For some experiments (DHE staining, immunohistochemistry, TUNEL, and EdU staining), whole-body perfusion and fixation were performed once the animal was under general anesthesia. Briefly, the mouse was anesthetized and placed in dorsal recumbency. A cut below the sternum was made, and then a V-shape incision was performed in the ribcage. The ribcage was grasped by the clamp to expose the heart. The venous infusion needle was inserted into the left ventricle, and the atrial appendage was immediately cut with scissors to allow blood to flow out. The peristaltic pump was turned on to perfuse saline into the whole body through the circulatory system. The switch from saline to 4% paraformaldehyde (PFA) was made when the fluid exiting the right atrium became clear and the liver color turned pale to red. Another 5 mL of 4% PFA was perfused when the mouse tail moved.

### 2.8. Immunohistochemistry

Immunohistochemistry staining was performed as we previously described [15], with some modifications. After washing with PBS 3 times, the slides were blocked by incubation with 10% goat serum in PBST for 1 h at room temperature (RT). Then the slides were incubated with primary antibodies (DCX: rabbit, Cell Signaling Technology 4604, Danvers, MA, USA dilution 1:500; p21: rabbit, Abcam ab188224, dilution 1:500) overnight at 4 °C. After washing for 3 times, the slides were then incubated with Biotin-SP AffiniPure Goat Anti-Rabbit lgG (H + L) secondary antibody (33103ES60, Yeasen, Shanghai, China dilution 1:200) for 2 h at RT, followed by incubation in an avidin–biotin–peroxidase solution (Vectastain Elite ABC kit; Vector Laboratories, Burlingame, CA, USA) for 1 h at RT. The color reaction was developed by incubation in diaminobenzidine (DAB Immunohistochemistry Color Development Kit, Sangon Biotech, Shanghai, China) in the presence of 0.01% H_2_O_2_ for 5 min at RT. Imaging was performed using an IX73 microscope (Olymupus, Tokyo, Japan). The number of DCX+ cells was counted using ImageJ software (1.50i, Laboratory for Optical and Computational Instrumentation, Madison, WI, USA).

### 2.9. Golgi Staining and Morphological Analysis of Dendrites and Spines 

Disruptions of hippocampal dentate granule neurons are closely correlated with stress-induced anxiety [16,17], and GC elevation is a primary endocrine response to stress. The hippocampus is well known as the principal target of GCs in the brain since it contains the greatest concentration of GC receptors. In order to know whether chronic exposure to CORT influences the structures of hippocampal neurons, the dendritic morphology and spine density of hippocampal dentate granule neurons were evaluated as described elsewhere [18,19] with some modifications. Golgi staining was performed using the Rapid GolgiStainTM Kit (FD NeuroTechnologies, Baltimore, MD, USA, PK401) according to the manufacturer’s instructions. Freshly dissected brains were immersed in a mixture of solutions A and B (1:1) and stored at room temperature in darkness for a duration of two weeks. Subsequently, the brains were transferred to solution C and maintained at 4 °C in the dark for 48 h. Following this, brain tissue was sectioned to a thickness of 100 µm, and the sections were stained in a mixture of solutions D, E, and H_2_O (1:1:2). Subsequently, the sections were subjected to standard staining protocols. The images of the dendrites were acquired with a 40× objective using an Olympus IX73 microscope (Tokyo, Japan), then a series of images at z-orientation were merged to reconstruct the image of dendritic trees by ImageJ software; the dendritic length and the branching points were analyzed using Sholl analysis with ring radius increments of 10 μm. Imaging of spines was carried out with a 100× oil-immersion objective using a Leica SP8 confocal microscope, and visible flanking spines were counted for 3–6 segments from each neuron. Dendrites and spines were evaluated for three mice in each group; for each hippocampus, three dentate granule neurons were randomly selected for subsequent analysis.

### 2.10. Western Blotting

The hippocampus of mice was quickly isolated, frozen using liquid nitrogen, and stored at −80 °C before processing. For Western blotting, hippocampal tissues were homogenized with RIPA buffer and protease inhibitor cocktails in a homogenizer at −50 °C (60 s). Lysates were then centrifuged (10,000× *g*, 5 min) before the supernatant was collected. Protein quantification was performed using a BCA kit, followed by denaturation of protein in sample loading buffer at 100 °C for 10 min. One hundred micrograms of total protein was loaded and separated in a 10% SDS-PAGE gel using running buffer. Then proteins were transferred to PVDF membrane in transfer buffer using a semi-dry transfer device (Bio-Rad, Berkeley, CA, USA), followed by blocking in TBST containing 5% non-fat milk for 1 h at RT. After washing with TBST three times, the membrane was incubated with primary antibodies (p21: rabbit, Abcam ab188224, dilution 1:500; β-actin: rabbit, Yeasen 30102ES40, dilution 1:1000) diluted in TBST containing 5% non-fat milk at 4 °C overnight on the shaker. After washing with TBST three times, the membranes were incubated with HRP-conjugated secondary antibodies at RT for 1 h. After washing for three times, the membranes were incubated in ECL working solution (Bio-Rad, USA) for 1 min and transferred to the Amersham Imager 600 (GE Healthcare, Chicago, IL, USA) immediately to detect the signals.

### 2.11. TUNEL

Terminal deoxynucleotidyl transferase dUTP nick end labeling (TUNEL) assay for detecting apoptosis was performed on frozen sections of the hippocampus using the TUNEL Alexa Fluor 640 kit (40308ES20, Yeasen Biotech, Shanghai, China) according to the manufacturer’s instructions. 

### 2.12. EdU Staining and Analysis

In order to visualize proliferative cells, mice received an intraperitoneal injection of EdU (5 mg/kg, MCE, HY-118411) dissolved in sterile PBS for a duration of 7 days, followed by euthanasia 8 days post-injection. Subsequently, transcardial perfusion and fixation were conducted once the animal was unconscious and completely insensible to pain. The brain was then isolated and immersed in 30% sucrose for dehydration. The thoroughly dehydrated tissues were embedded in O.C.T. compound and stored at −80 °C until use. The frozen tissue blocks were sectioned at a thickness of 20 μm using a cryostat (Leica CM 1950). EdU staining was performed by using the Click-iT Alexa Fluor 488 Imaging Kit (C10337, Thermo Fisher Scientific, Waltham, MA, USA) according to the manufacturer’s instructions. EdU-positive cells were quantified in the brain sections (20 μm thickness, four-section interval) throughout the entire hippocampus using ImageJ software (1.50i, Laboratory for Optical and Computational Instrumentation, Madison, WI, USA).

### 2.13. mRNA Extraction and Real-Time Quantitative PCR

The hippocampus was rapidly isolated and immediately transferred into TRIzol. Subsequently, the hippocampus was homogenized in a homogenizer at −50 °C (65 Hz, 60 s). The resultant homogenate was then treated with DNase I to remove any residual genomic DNA. The total RNA was diluted in RNAase-free water, and 1 μg of RNA was reverse-transcribed into complementary DNA (cDNA) using the PrimeScriptTMRT reagent kit (TAKARA, RR037A). Quantitative real-time PCR (qPCR) was carried out using the SYBR Green Mix from Bio-Rad, and the qPCR reactions were conducted using a Bio-Rad CFX96 Thermal Cycler. To normalize the expression levels of the target genes and account for potential variations among distinct samples, the expression level was referenced to the housekeeping gene β-actin.

### 2.14. Transcriptional Analysis of Hippocampus by RNA-Sequencing (RNA-Seq)

The mouse was euthanized, and the hippocampus was carefully isolated and preserved at −80 °C for subsequent analysis. The frozen tissue samples (50 mg/sample) were sent to the company (Majorbio Bio-Pharm Technology Co., Ltd., Shanghai, China) for RNA extraction and RNA-seq analysis. In summary, total RNA was extracted from each experimental group with three biological replicates. The RNA-seq transcriptome library was generated using 1 μg of total RNA and the TruSeqTM RNA sample preparation Kit from Illumina (San Diego, CA, USA). Following library preparation, the paired-end RNA-seq library underwent sequencing utilizing the Illumina NovaSeq 6000 sequencer, generating reads of 2 × 150 bp in length. Subsequently, the raw paired-end reads were processed for trimming and quality control using the fastp tool (https://github.com/OpenGene/fastp). The processed reads were then aligned to the reference genome in orientation mode, employing the HISAT2 software. The mapped reads for each sample were assembled using StringTie through a reference-based approach. The expression levels of individual transcripts were calculated using the transcripts per million reads (TPM) method. Transcript abundances were quantified utilizing the RSEM software tool. Differential expression analysis was conducted utilizing DESeq2, with a threshold of log2FC > 0 and adjusted *p*-value < 0.05. KEGG pathway analysis was performed employing KOBAS (http://kobas.cbi.pku.edu.cn/home.do).

DEGs were mapped to the search tool (https://string-db.org) for retrieval of interacting genes to acquire protein–protein interaction (PPI) networks. Visualization of the PPI network was performed by Cytoscape software (3.9.1). CytoHubba, a plugin in Cytoscape, was used for ranking nodes and screening hub genes in the network. GSEA analysis was performed on the Majorbio I-Sanger Cloud platform (https://cloud.majorbio.com/).

### 2.15. Statistical Analysis

Statistical analysis was performed using GraphPad PRISM 8 (GraphPad Software, San Diego, CA, USA). Unpaired student’s *t*-test was used to compare the difference between two groups, whereas a two-way ANOVA was used to compare the difference among three or more groups. All data were expressed as the mean ± the standard error of the mean (SEM). * *p* < 0.05 and ** *p* < 0.01 were considered to be significant.

## 3. Results

### 3.1. CORT Induced Anxiety- and Depressive-like Behaviors with Reduced Hippocampal Neurogenesis and Dendritic Atrophy

In order to evaluate the influence of prolonged corticosterone (CORT) administration on mice hippocampus and behaviors, mice were exposed to CORT for 36 days (Figure 1A). The total distance traveled in the open field arena was comparable between the two groups of mice (Figure 1B,C), indicating that locomotor activity was not impaired by CORT treatment. CORT-treated mice exhibited reduced activity and decreased time spent exploring the center of the open field (Figure 1D,E) or the open arm of the elevated plus maze (Figure 1F–H), indicating an anxiety-like state of mice. Moreover, the CORT-treated mice showed a depression-like condition as measured by increased immobility in FST and TST (Figure 1I,J). The hippocampus is well known as the principal target of GCs in the brain since it contains the greatest concentration of GC receptors, and animal studies reported that chronic exposure to CORT can cause damage to the hippocampus [7,10]. Therefore, hippocampal histological analysis was conducted. The number of EdU-positive and doublecortin (DCX)-positive cells was significantly decreased in CORT-treated mice (Figure 1K–N), indicating that CORT treatment led to the reduced proliferation of neural stem cells and newborn immature neurons in the dentate gyrus (DG) of the hippocampus, respectively. Contrary to the initial belief that neuron loss predominantly contributes to hippocampal volume reduction, recent studies suggest that dendritic and synaptic alterations are more likely to be responsible for hippocampal shrinkage [20]. Therefore, we explored the morphology of dendritic spines: CORT treatment significantly reduced the complexity of dendritic branching (Figure 1O–R) and spine density (Figure 1S,T). Altogether, prolonged CORT treatment led to depressive- and anxiety-like behaviors in mice, accompanied by reduced hippocampal neurogenesis and dendritic atrophy.

### 3.2. Identification of the Hub Gene p21 Interacting with Oxidative Stress-Associated Differentially Expressed Genes (DEGs) in CORT-Treated Hippocampus by RNA-Sequencing (RNA-Seq)

To further uncover the underling mechanism by which CORT caused hippocampal alterations and related behavior deficits, we performed RNA-seq analysis of the mice hippocampus. The samples of the two groups were remarkably divided into two populations based on the principal component analysis (PCA) (Figure 2A). Approximately 2404 DEGs (genes with >1-fold change in expression, *p* < 0.05), including 1183 down-regulated and 1221 up-regulated genes, were identified in the CORT-treated group (Figure 2B). Kyoto Encyclopedia of Genes and Genomes (KEGG) analysis was performed to investigate the functional categories of the up-regulated DEGs, and “cell growth and death” pathway was identified to be the top 2 enriched pathway (Figure 2C), indicating dysregulated cell proliferation in the CORT-treated group. In order to find the key genes associated with CORT-induced abnormal proliferation, we proposed a gene set enrichment analysis (GSEA) approach to analyze (protein–protein interaction) PPI networks. GSEA exposed that 258 out of the 2404 DEGs were highly enriched in 40 gene ontology (GO) pathways, among which “intrinsic apoptotic signaling pathway by p53 class mediator” was the top 1 gene set ranked by normalized enrichment score (NES) (Figure 2D). Then the 258 DEGs in 40 GO pathways were analyzed by PPI networks (Figure 2E, Appendix A): The up-regulated *Cdkn1a* (*p21*), which belonged to the “intrinsic apoptotic signaling pathway by p53 class mediator” pathway, was one of the top 20 hub genes, indicating that abnormally elevated *p21* might be the key factor responsible for CORT-induced hippocampal alterations. Moreover, PPI networks exposed the interaction of *p21* and oxidative stress-related genes (enlarged inset, Figure 2E). Quantitative polymerase chain reaction (q-PCR) validated the significant up-regulation of *p21* and oxidative stress-associated genes (*Parp1*, *Didt4*, *Trp53inp1*, *Pmaip1*, *Akt2*) in the CORT-treated group (Figure 2F). Western blotting (Figure 2G,H) and immunohistochemistry (Figure 2I) demonstrated elevated p21 expression in the hippocampus of CORT-treated mice. Moreover, ROS accumulation was identified in the DG of CORT-treated mice (Figure 2J,K). According to the results above, we speculated that prolonged exposure to CORT elevated *p21* expression and consequently activated oxidative stress-associated pathways and ROS accumulation, thus leading to hippocampal alterations and associated behavioral deficits. 

### 3.3. p21-Overexpression Led to ROS Accumulation, Reduction in Hippocampal Neurogenesis and Dendritic Branching, and Anxiety-like Behaviors

In order to validate the role of *p21*, we conducted a *p21* overexpression (*p21*-OE) experiment by bilateral stereotaxic injection of recombinant AAV (rAAV) into the DG of mice (Figure 3A,B). Four weeks after the stereotactic surgery, the two groups of mice showed no statistical difference in total distance traveled in the open field arena (Appendix A), indicating that locomotor activity was not influenced by *p21*-OE. Mice of the *p21*-OE group exhibited anxiety-like behaviors, as evidenced by a significant decrease in their tendency to explore the open arms (Figure 3C–E). Forty-three days after AAV injection, histological analysis showed the expression of mNeonGreen in DG area of mice, indicating that AAVs-mediated gene transfer can achieve efficient long-term expression in the hippocampus (Figure 3F). We validated the remarkable up-regulation of *p21* expression in the hippocampus of the *p21*-OE group by qPCR (Figure 3G) and Western blotting (Figure 3H,I). *p21*-OE mice also showed ROS accumulation in the DG area (Figure 3J), similar to what we observed in CORT-treated mice (Figure 2J,K). The number of EdU-positive and DCX+ cells significantly decreased in *p21*-OE mice (Figure 3K,L), indicating reduced hippocampal neurogenesis due to *p21* overexpression. Moreover, *p21*-OE mice showed reduced complexity of dendritic branching (Figure 3M–P) and spine density of granule neurons (Figure 3Q,R). This result suggested that *p21*-overexpression in hippocampus almost recapitulated CORT-induced behavior deficits and histological alterations. 

We subsequently investigated whether knock-down of *p21* was sufficient to recover the phenotypes caused by CORT treatment. However, specific knockdown of *p21* in the hippocampus (Appendix A) could not recover the CORT-induced phenotypes but instead led to apoptosis of granular neurons (Appendix A).

### 3.4. Inhibition of ROS Partly Rescued Anxiety-like Behaviors Accompanied by Hippocampal Alterations Induced by p21-Overexpression

Elevated ROS is considered to cause sustained suppression of hippocampal neurogenesis [21], and we also identified that *p21* overexpression led to ROS accumulation (Figure 3J) and reduced hippocampal neurogenesis in the DG area (Figure 3K,L). This prompted us to speculate whether inhibition of ROS could prevent *p21*-OE-induced hippocampal alterations and behavioral deficits. *P21* overexpression was conducted by bilateral stereotaxic injection of AAV-Cdkn1a-3×Flag-mNeonGreen, while antioxidant N-acetyl cysteine (NAC) was treated to mice in drinking water to scavenge ROS (Figure 4A). NAC treatment inhibited *p21*-OE-induced anxiety-like behaviors (Figure 4B–D). Histological analysis confirmed that antioxidant NAC treatment partly inhibited *p21*-OE-mediated ROS production (Figure 4E,F) and attenuated *p21*-OE-induced hippocampal alterations, including the reduction in hippocampal neurogenesis (Figure 4G–J) and dendritic atrophy (Figure 4K–P). This result suggested that *p21* overexpression directly induced ROS accumulation in the hippocampus, thus leading to hippocampal histological alterations and associated behavioral deficits.

### 3.5. Inhibition of ROS Partially Rescued CORT-Induced Anxiety-like Behaviors Accompanied with Hippocampal Alterations

According to the results above, we proposed a hypothesis: Chronic CORT treatment elevated *p21* expression, which contributed to ROS accumulation, consequently leading to hippocampus alterations and associated behavioral deficits (Figure 2). In order to prove this hypothesis, we investigated the influence of antioxidant NAC on CORT-induced alterations (Figure 5A). Treatment with NAC inhibited CORT-induced anxiety-like behaviors (Figure 5B–D) without influencing depressive-like behaviors (Appendix A). Histological analysis confirmed that NAC treatment partially inhibited CORT-induced ROS production (Figure 5E,F) and mitigated CORT-induced hippocampal alterations, including the reduction in hippocampal neurogenesis (Figure 5G–J) and dendritic atrophy (Figure 5K–P). These results strengthened our conclusion: long-term exposure to CORT up-regulated the *P21*-associated pathways, which activated the downstream oxidative stress-related pathways, causing ROS accumulation in the hippocampus; the excessive ROS eventually led to reduced hippocampal neurogenesis and dendritic atrophy, which subsequently contributed to abnormal behavioral alterations.

## 4. Discussion

This study investigates the mechanism by which chronic exposure to CORT induces hippocampal alterations and the associated behavioral deficits: (1) Chronic exposure to CORT induces anxiety- and depressive-like behaviors, coupled with diminished hippocampal neurogenesis and dendritic atrophy; (2) by employing RNA-seq, we identify oxidative-associated DEGs, pinpointing *p21* as a key hub gene associated with CORT-induced hippocampal proliferation alterations; (3) overexpression of *p21* is found to mostly recapitulate CORT-induced alterations, including anxiety-like behaviors, reduced hippocampal neurogenesis and dendritic atrophy, accompanied by ROS accumulation; (4) notably, we find that inhibition of ROS presents a promising avenue for mitigating anxiety-like behaviors and hippocampal alterations induced by *p21* overexpression; (5) furthermore, targeting ROS could also hold potential in partially ameliorating CORT-induced anxiety-like behaviors, concurrently addressing the associated hippocampal alterations.

The observed induction of anxiety- and depressive-like behaviors following chronic exposure to CORT (Figure 1A–J) aligns with the existing knowledge regarding the role of CORT as a well-validated pharmacological factor in regulating mood and behaviors [22]. Consistent with the previous studies [23,24,25,26], we found that prolonged exposure to CORT reduced hippocampal cell proliferation, decreased the complexity of dendritic branching and spine density of hippocampal granular neurons (Figure 1K–T). The histological alterations of hippocampus (Figure 1K–T) are particularly concerning given the hippocampus’s critical role in emotional regulation: the hippocampus is located in the medial temporal lobe, lying posterior to the amygdala, a center responsible for emotional memory recalling and regulation; scientists observed an augmentation in the functional connectivity between the hippocampus and the amygdala during emotional regulation and the retrieval of positive memories [27]; more importantly, researchers also confirmed that the hippocampus is involved in emotional regulation by using real-time functional magnetic resonance imaging (fMRI) technology [28,29], suggesting the potential clinical application of hippocampal regulation in treating mental disorders, including addiction, anxiety, and depression. The hippocampal alterations observed in our study suggest a potential mechanism through which exposure to CORT disrupts emotional processes. Consequently, our focus was directed towards investigating the transcriptomic alterations in the CORT-treated hippocampus, trying to elucidate the plausible molecular mechanism by which CORT induces hippocampal histological alterations and the related emotional disorders. 

Understanding the interactions of *p21* with oxidative stress-related DEGs is crucial in unraveling the intricate molecular pathways that mediate the impact of CORT on hippocampal neurogenesis and hippocampus-mediated mood regulation. In this study, *p21*-overexpression partially recapitulated CORT-induced phenotypes, including ROS accumulation, reduction in hippocampal neurogenesis and dendritic branching, and anxiety-like behaviors (Figure 3); and antioxidants partially rescued the *p21*-overexpression-induced phenotypes (Figure 4), indicating that ROS accumulation is a direct consequence of *p21* up-regulation. Consistent with our study, Macip et al. demonstrated that tet-regulatable *p21* expression increased intracellular ROS both in normal fibroblasts and in p53-negative cancer cells, and antioxidant NAC can block ROS accumulation in response to tet-regulatable *p21* [30]. Actually, the ROS accumulation in response to *p21* up-regulation can be explained by the following mechanisms: (1) persistent up-regulation of the cell cycle checkpoint gene *p21* leads to mitochondrial impairment and subsequent generation of ROS through a sequential signaling cascade GADD45-p38MAPK-GRB2-TGFBR2-TGFβ [31]; (2) ROS-induced DNA damage causes p53 activation [32], which in turn up-regulates its downstream target *p21*, thus establishing a reinforcing feedback loop constituted by *p21*-ROS-P53.

It is noteworthy that antioxidant treatment partially mitigated the hippocampal damage and anxiety-like behaviors induced by *p21*-OE and prolonged exposure to CORT, as illustrated in Figure 4 and Figure 5. Therefore, in addition to the elucidated ROS pathway in this study, there might be other molecular events involved in CORT-induced-*p21* up-regulation, contributing to the modulation of hippocampal alterations and resultant behavioral deficits. Several studies have elucidated the pivotal role of *p21* in regulating hippocampal neurogenesis in response to diverse stimuli: Zonis et al. demonstrated that *p21* up-regulation in response to acute inflammation restrained neuronal progenitor proliferation in mice hippocampus [33]; Pechnick et al. reported that antidepressants could promote hippocampal neurogenesis by inhibiting *p21* expression [34], underscoring the significance of *p21* in modulating hippocampal neurogenesis. Notably, *p21*, as a cyclin-dependent kinase inhibitor, induces cell cycle arrest in response to various stimuli. Therefore, CORT-induced *p21* up-regulation might directly lead to cell cycle arrest in neural progenitors within the hippocampus, which cannot be rescued by antioxidant treatment.

All of these findings strongly suggest that ROS accumulation is responsible for the reduced hippocampal neurogenesis and dendritic atrophy initiated by CORT-mediated up-regulation of *p21*. In the adult hippocampus, the formation of new neurons (neurogenesis) occurs throughout life and is critical for emotional memory recalling and regulation. In turn, deficits in neurogenesis over time may compromise hippocampal function, gradually leading to mood disorders. Actually, impaired hippocampal neurogenesis is related to the development of anxiety-like behaviors. For example, young rats that experienced childhood neglect showed anxiety in adulthood and decreased AHN [35]. Transgenic animals with impaired AHN induced by overexpression of the pro-apoptotic protein Bax exhibited anxiety-like behaviors [36]. Therefore, ROS-induced oxidative stress can disrupt the delicate balance of hippocampal neurogenesis, potentially leading to impaired hippocampus structure and its function in mood regulation.

## 5. Conclusions

In conclusion, the identification of *p21* as a hub gene interacting with oxidative stress-associated DEGs sheds light on potential therapeutic targets and signaling pathways that could be modulated to mitigate the adverse effects of chronic stress or prolonged CORT exposure (such as in Cushing’s syndrome-endogenous GCs or when using high doses of exogenous GCs) on hippocampus structure and function.

## Figures and Tables

**Figure 1 biomolecules-14-00268-f001:**
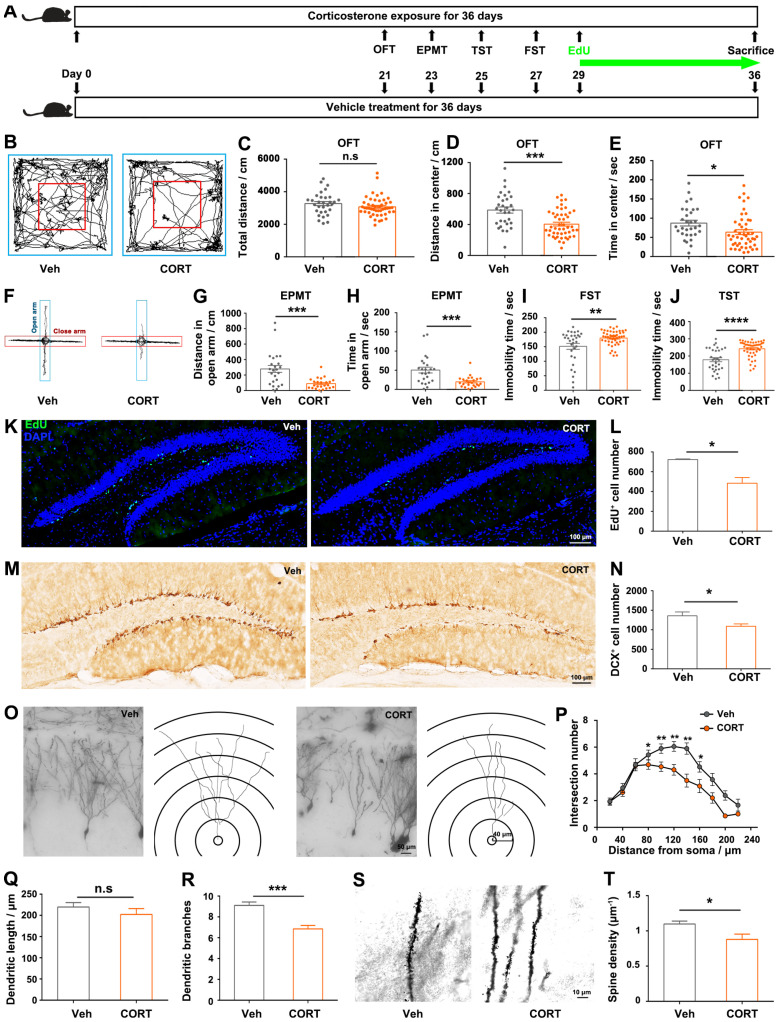
Chronic CORT treatment induced anxiety- and depression-like behaviors, accompanied with reduced neurogenesis and dendritic atrophy in hippocampus. (**A**) Study design. (**B**) Representative trajectory diagram of the mice in the open field test. Red box indicates the center zone of the arena. (**C**) Total distance traveled by mice in the whole area of the apparatus. Distance traveled (**D**) and time spent (**E**) by mice in the central zone of the open field. (**F**) Representative trajectory diagrams of the mice in the elevated plus maze test. Distance traveled (**G**) and time spent (**H**) in the open arm by mice. Immobility time of the mice in the forced swimming test (**I**) and tail suspension test (**J**). Representative images (**K**) and quantification (**L**) of EdU staining within the dentate gyrus (DG) area. Representative images (**M**) and quantification (**N**) of DCX staining within the DG area. (**O**) Representative images and tracings of the granule cell in DG area. Quantification of intersection number (**P**), dendritic length (**Q**) and branch number (**R**) of granule cells. (**S**) Representative images of dendritic spines of granule cell in DG area. (**T**) Quantitative analysis of spine density of the granular cell dendrites. Data are presented as mean ± SEM, n = 31 per group for behavior tests; otherwise, n = 3 per group. Statistical comparisons were performed by Student’s *t* test. * *p* < 0.05, ** *p* < 0.01, *** *p* < 0.001, **** *p* < 0.0001, n.s.: no significance. EPMT: elevated plus maze test; OFT: open field test; TST: tail suspension test; FST: forced swimming test.

**Figure 2 biomolecules-14-00268-f002:**
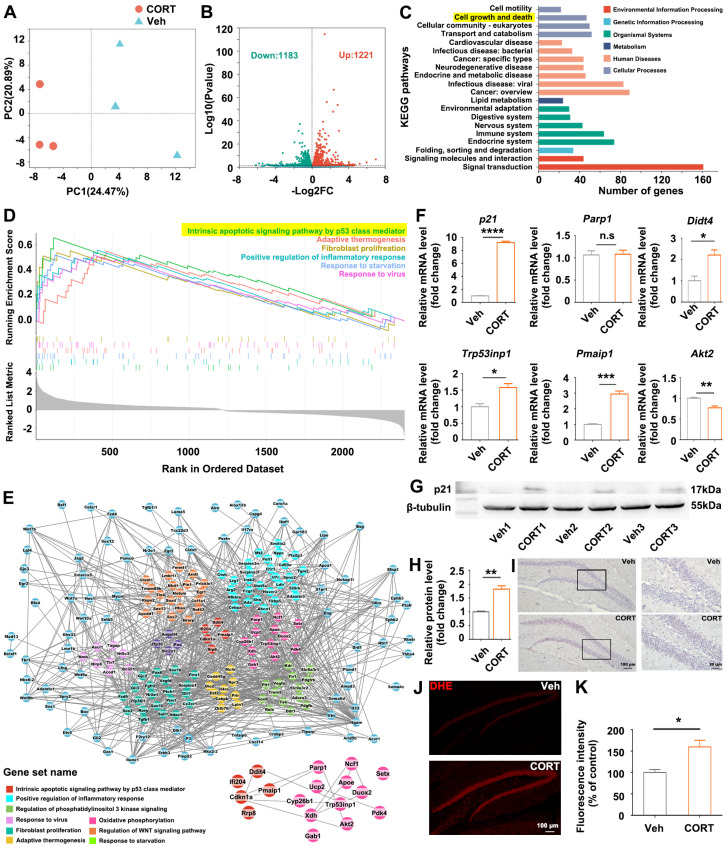
Identification of key pathways and hub genes contributing to CORT-induced abnormal hippocampal neurogenesis. (**A**) Principal component analysis (PCA) of the transcriptome of the hippocampus samples collected from the mice administrated CORT or vehicle. (**B**) Volcano plots of the DEGs for hippocampus of the 2 groups. The up-regulated DEGs are presented by red dots, and the down-regulated DEGs are presented by green dots. (**C**) The functional classification of the DEGs based on KEGG pathway analysis. (**D**) GSEA enrichment score curves of the top 6 GO pathways. The top panel shows the enrichment scores of each pathway. The middle vertical bars display the position of each gene. The bottom panel indicates the ranking metric scores. (**E**) Protein–protein interaction (PPI) network of the important genes. The grey lines indicated the associations between the genes. (**F**) The mRNA expression levels of *p21* and the oxidative stress-associated genes. Western blot analysis (**G**) and quantification (**H**) of p21 protein expression level. (**I**) Representative images of p21 immunostaining in hippocampus of mice. Representative images (**J**) and quantification (**K**) of DHE staining in hippocampus of mice. Data are presented as mean ± SEM, n = 3 per group, statistical comparisons were performed by Student’s *t* test. * *p* < 0.05, ** *p* < 0.01, *** *p* < 0.001, **** *p* < 0.0001, n.s.: no significance.

**Figure 3 biomolecules-14-00268-f003:**
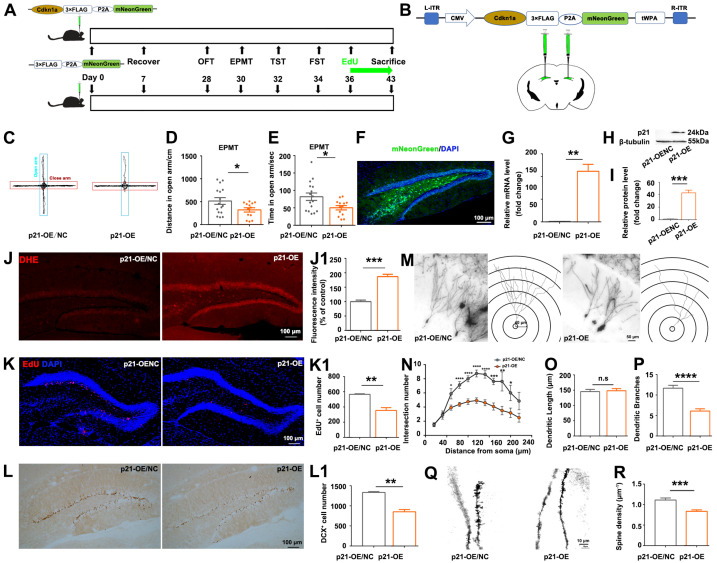
Recapitulation of CORT-induced behavior deficits and histological alterations by *p21* overexpression. (**A**) Study design. (**B**) Schematic of stereotaxic injection of AAV vectors into the bilateral DG areas of mice brain. (**C**) Representative trajectory diagrams of the mice in the elevated plus maze test. Distance traveled (**D**) and time spent (**E**) in the open arm by mice. (**F**) Anatomical spread of AAV vectors visualized by mNeonGreen fluorescence. (**G**) Analysis of *p21* mRNA expression level. Western blot analysis (**H**) and quantification (**I**) of p21 protein expression level. Representative images (**J**) and quantification (**J1**) of DHE staining within the dentate gyrus (DG) area. Representative images (**K**) and quantification (**K1**) of EdU staining within the dentate gyrus (DG) area. Representative images (**L**) and quantification (**L1**) of DCX staining within the dentate gyrus (DG) area. (**M**) Representative images and tracings of the granule cell in DG area. Quantification of intersection number (**N**), dendritic length (**O**), and branch number (**P**) of granule cells. (**Q**) Representative images of dendritic spines of granule cells in DG area. (**R**) Quantitative analysis of spine density of the granular cell dendrites. Data are presented as mean ± SEM, n = 16 per group for behavior tests; otherwise, n = 3 per group. Statistical comparisons were performed by Student’s t test. * *p* < 0.05, ** *p* < 0.01, *** *p* < 0.001, **** *p* < 0.0001, n.s.: no significance. EPMT, elevated plus maze test.

**Figure 4 biomolecules-14-00268-f004:**
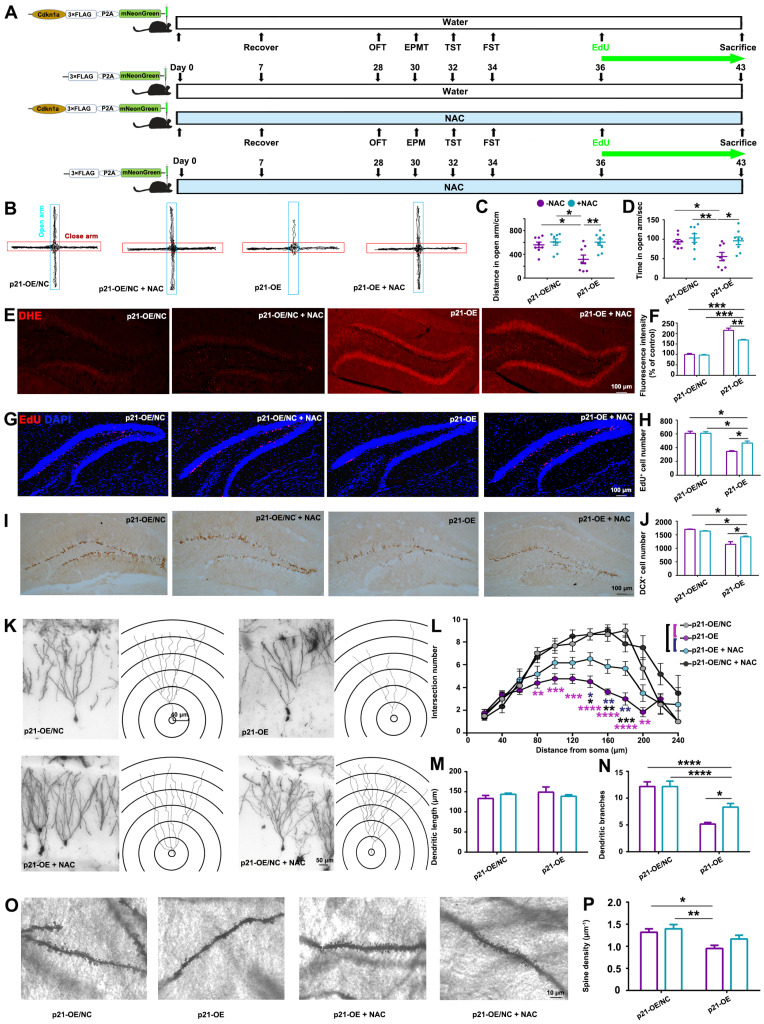
Antioxidant treatment partially rescued *p21*-OE-induced phenotypes. (**A**) Study design. (**B**) Representative trajectory diagrams of the mice in the elevated plus maze test. Distance traveled (**C**) and time spent (**D**) in the open arm by mice. Representative images (**E**) and quantification (**F**) of DHE staining within the DG area. Representative images (**G**) and quantification (**H**) of EdU staining within the DG area. Representative images (**I**) and quantification (**J**) of DCX staining within the DG area. (**K**) Representative images and tracings of the granule cell in DG area. Quantification of intersection number (**L**), dendritic length (**M**) and branch number (**N**) of granule cells. (**O**) Representative images of dendritic spines of granule cells in DG area. (**P**) Quantitative analysis of spine density of the granular cell dendrites. Data are presented as mean ± SEM, n = 8 per group for behavior tests; otherwise, n = 3 per group. Statistical comparisons were performed by one-way ANOVA. * *p* < 0.05, ** *p* < 0.01, *** *p* < 0.001, **** *p* < 0.0001, n.s.: no significance. EPMT, elevated plus maze test.

**Figure 5 biomolecules-14-00268-f005:**
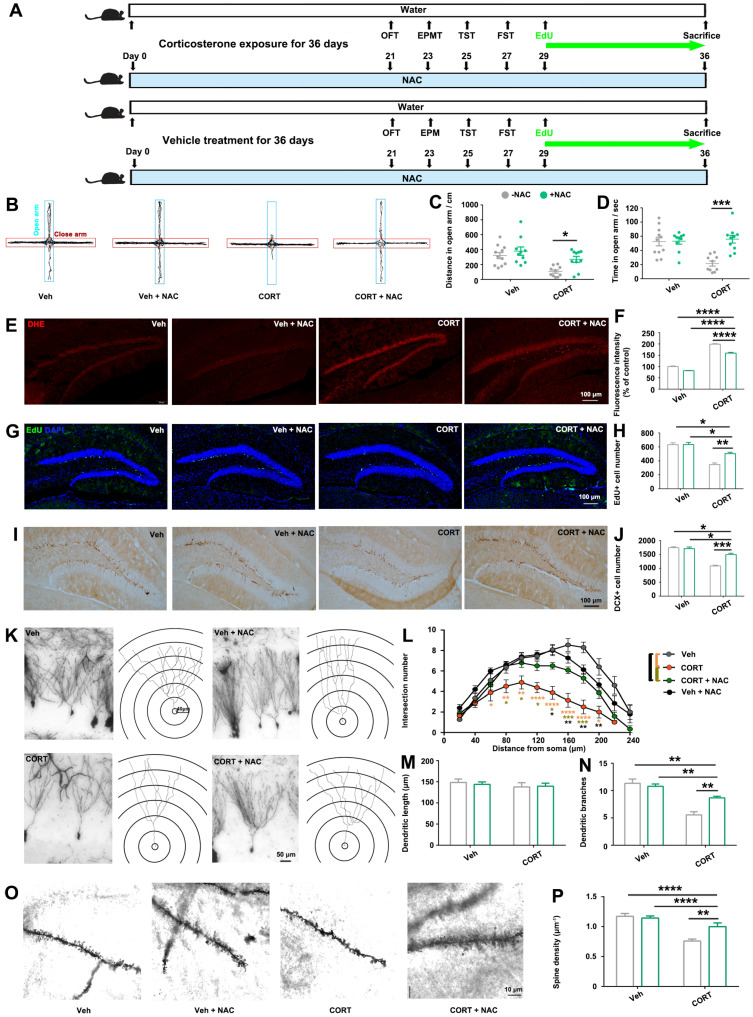
Antioxidant treatment partially rescued CORT-induced phenotypes. (**A**) Study design. (**B**) Representative trajectory diagrams of the mice in the elevated plus maze test. Distance traveled (**C**) and time spent (**D**) in the open arm by mice. Representative images (**E**) and quantification (**F**) of DHE staining within the DG area. Representative images (**G**) and quantification (**H**) of EdU staining within the DG area. Representative images (**I**) and quantification (**J**) of DCX staining within the DG area. (**K**) Representative images and tracings of the granule cell in DG area. Quantification of intersection number (**L**), dendritic length (**M**) and branch number (**N**) of granule cells. (**O**) Representative images of dendritic spines of granule cells in DG area. (**P**) Quantitative analysis of spine density of the granular cell dendrites. Data are presented as mean ± SEM, n = 12 per group for behavior tests; otherwise, n = 3 per group. Statistical comparisons were performed by one-way ANOVA. * *p* < 0.05, ** *p* < 0.01, *** *p* < 0.001, **** *p* < 0.0001, n.s.: no significance. EPMT, elevated plus maze test.

## Data Availability

RNA-seq data have been deposited in the Gene Expression Omnibus (GEO) database. The accession number for the RNA-seq data reported in this paper is: GSE244413. The following secure token has been created to allow review of GSE244413 while it remains in private status: snmfwkqifzcnxix.

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
