# Peer review of "Corticosterone Impairs Hippocampal Neurogenesis and Behaviors through p21-Mediated ROS Accumulation"

_biomolecules, 2024, doi:10.3390/biom14030268_

Round 1

Reviewer 1 Report

Comments and Suggestions for Authors

This study by Wang explores how chronic exposure to CORT leads to hippocampal changes and related behavioral issues. The findings include: (1) CORT exposure causes anxiety and depression-like behaviors, along with reduced hippocampal neurogenesis and dendritic atrophy; (2) Using RNA-seq, the authors have identified oxidative-associated genes, highlighting p21 as a crucial hub gene linked to CORT-induced hippocampal proliferation changes; (3) Overexpression of p21 replicates CORT-induced alterations, including anxiety-like behaviors, decreased hippocampal neurogenesis, and dendritic atrophy, along with ROS accumulation; (4) Inhibiting ROS shows promise in alleviating anxiety-like behaviors and hippocampal changes induced by p21 overexpression; (5) Targeting ROS may also partially improve CORT-induced anxiety-like behaviors while addressing associated hippocampal alterations.

Collectively, the manuscript is well written and presented. However, I have some suggestions that might need to be addressed by the authors.

·       The introduction is lacking the aim of the study. The authors have nicely written an introduction about the relevance of glucocorticoids in the hippocampus in relation to anxiety disorder, but the rationale of why they have designed these series of experiments is not clear to me.

·       As the study includes many experiments, in the material and methods section, I would suggest naming the different section as Exp1: XX, Exp2: XX (i.e Experiment 1: Prolonged exposure to corticosterone in mice, Experiment 2: Manipulation of p21 expression in hippocampal dentate gyrus by stereotaxic injection of AAV vectors, etc) and follow the same name in the results section as it would be easier to understand.

·       Were all the mice perfused at the end of the experimental design? It is not clear to me, if they were, how molecular techniques such as qPCR, Western Blot were carried out in perfused tissue. In the methods section, in the Golgi protocol it specifies “freshly dissected brains”  but it is not clear in the other techniques and mice from the same experiment are used in a wide range of molecular assays.

·       As microglia can become activated by challenges as stressful events, and these microglia produce reactive oxygen species, would be interesting in future experiments flowing this investigation, whether p21 overexpressed- animals have also an increase in this cell type.

·       Additionally, in order to further test the conclusion of the authors “All of these findings suggest strongly that ROS accumulation, is responsible for…” would be nice to somehow measure 2-OH-E fluorescence, a marker for ROS activity.

Author Response

General comment:

This study by Wang explores how chronic exposure to CORT leads to hippocampal changes and related behavioral issues. The findings include: (1) CORT exposure causes anxiety and depression-like behaviors, along with reduced hippocampal neurogenesis and dendritic atrophy; (2) Using RNA-seq, the authors have identified oxidative-associated genes, highlighting p21 as a crucial hub gene linked to CORT-induced hippocampal proliferation changes; (3) Overexpression of p21 replicates CORT-induced alterations, including anxiety-like behaviors, decreased hippocampal neurogenesis, and dendritic atrophy, along with ROS accumulation; (4) Inhibiting ROS shows promise in alleviating anxiety-like behaviors and hippocampal changes induced by p21 overexpression; (5) Targeting ROS may also partially improve CORT-induced anxiety-like behaviors while addressing associated hippocampal alterations.

Collectively, the manuscript is well written and presented. However, I have some suggestions that might need to be addressed by the authors.

General response:

We sincerely appreciate the reviewer for taking the time to review our manuscript. The valuable comments are helpful for us to improve this manuscript. We have carefully reviewed the comments and have revised the manuscript accordingly. Please see below, in blue, for a point-by-point response to the comments. All the modifications in the manuscript are shown in red.

Comment 1: The introduction is lacking the aim of the study. The authors have nicely written an introduction about the relevance of glucocorticoids in the hippocampus in relation to anxiety disorder, but the rationale of why they have designed these series of experiments is not clear to me.

Response 1: Thanks for the comment, we have carefully revised the “Introduction section” to address the concerns raised regarding the aim of the study. In the revised version, we have provided a more explicit rationale for the series of experiments conducted.

The added content for “Introduction” section can be found in the revised manuscript.

The revised content in the manuscript is as follows:

Introduction: Page 2, line 64-72.

Comment 2: As the study includes many experiments, in the material and methods section, I would suggest naming the different section as Exp1: XX, Exp2: XX (i.e Experiment 1: Prolonged exposure to corticosterone in mice, Experiment 2: Manipulation of p21 expression in hippocampal dentate gyrus by stereotaxic injection of AAV vectors, etc) and follow the same name in the results section as it would be easier to understand.

Response 2: In response to your recommendation, we have added serial numbers to the different experiments for improved clarity and understanding. However, to adhere to the journal template format requirements, we have chosen to label the sections as follows:

2.1. Housing of animals

2.2. Prolonged exposure to corticosterone in mice

…etc

We believe that this modification maintains clarity while aligning with the prescribed format of the journal template. We hope this meets your expectations and enhances the overall readability of the manuscript.

The revised content in the manuscript is as follows:

Materials and Methods:

Page 2, line 84, 91, 98-99;

Page 3, line 116, 127, 135, 141, 146, 151;

Page 4, line 157, 165, 177, 191;

Page 5, line 214, 231, 236, 248, 259;

Page 6, line 283.

Comment 3: Were all the mice perfused at the end of the experimental design? It is not clear to me, if they were, how molecular techniques such as qPCR, Western Blot were carried out in perfused tissue. In the methods section, in the Golgi protocol it specifies “freshly dissected brains”, but it is not clear in the other techniques and mice from the same experiment are used in a wide range of molecular assays.

Response 3: Regarding your concern about the perfusion of mice, we would like to clarify that not all mice were perfused. In specific experiments, we provided information on whether the mice were perfused or not. Specifically, transcardial perfusion was only performed for 4 experiments: DHE staining, immunohistochemistry, TUNEL, EdU staining. And we have made the necessary updates to reflect this in the “Materials and Methods” section.

The revised content in the manuscript is as follows:

Materials and Methods:

Page 4, line 165-176;

Comment 4: As microglia can become activated by challenges as stressful events, and these microglia produce reactive oxygen species, would be interesting in future experiments flowing this investigation, whether p21 overexpressed- animals have also an increase in this cell type.

Response 4: This is a good suggestion, we will take your recommendation into consideration for our future studies and plan to explore the relationship between p21 overexpression and microglia activation, specifically examining whether there is an increase in this cell type and associated ROS production. This additional investigation will contribute to a more comprehensive understanding of the mechanisms involved and the broader effects of p21 overexpression in the context of chronic stress-induced neuropathological changes.

Comment 5: Additionally, in order to further test the conclusion of the authors “All of these findings suggest strongly that ROS accumulation, is responsible for…” would be nice to somehow measure 2-OH-E fluorescence, a marker for ROS activity.

Response 5: We appreciate the thoughtful suggestion from the reviewer. Dihydroethidium (DHE) is a commonly used fluorogenic dye for the detection of reactive oxygen species (ROS) in cells and tissues. When DHE is oxidized by ROS, it forms a fluorescent product (2-hydroxyethidium, also known as 2-OH-E+) that can be visualized and quantified.

Actually, in the original manuscript, we conducted DHE staining to assess ROS accumulation, and quantified the fluorescence signal associated with 2-hydroxyethidium (2-OH-E+), a marker for ROS activity. The results of the DHE staining (red fluorescence of 2-OH-E+) and quantification were presented in the figures of the manuscript (Fig. 2J-K; Fig. 3J-J1; Fig. 4E-F; Fig. 5E-F).

Please also see the attachment.

Reviewer 2 Report

Comments and Suggestions for Authors

The paper is well written and the study is well designed.  However it should be improved to be published.

1. The sequences of the primers for r-t qPCR should be provided in supplementary materials.

2. When it is possible, individual data should be plotted in histograms (stat plots) to reveal the distribution of data. As some histograms in present version of manuscript are shown individual data (e.g. Fig 3 D,E; Fig 4 C,D; Fig 5C,D), it is strange that others (e.g. Fig 4M,N and many others) are not. Why?

The article could be published after minor improvements.

Author Response

General comment:

The paper is well written and the study is well designed.  However it should be improved to be published.

General response:

We thank the reviewer for the professional comments on our manuscript. We really appreciate you taking the time to review this paper. We have carefully revised the manuscript accordingly. The point-by-point responses to the reviewer’s comments are as follows in blue. All the modifications in the manuscript are shown in red.

Comment 1: The sequences of the primers for r-t qPCR should be provided in supplementary materials.

Response 1: Thanks for the comment, we revised the content accordingly, please find the information for the sequences of the primers for r-t qPCR in supplementary table2 (Supporting information, PDF).

The revised content in the manuscript is as follows:

Supporting information: Supplementary file, page 9, Supplementary table2.

Comment 2: When it is possible, individual data should be plotted in histograms (stat plots) to reveal the distribution of data. As some histograms in present version of manuscript are shown individual data (e.g. Fig 3 D,E; Fig 4 C,D; Fig 5C,D), it is strange that others (e.g. Fig 4M,N and many others) are not. Why?

Response 2: Thanks for the question. In the present study, individual data were only shown for animal behavioral analysis (Fig 3 D,E; Fig 4 C,D; Fig 5C,D). However, for other types of analyses, such as qPCR, quantification of Western blotting bands and immunofluorescence, bar graphs were used to represent data. And the reasons are listed as following:

For behavioral Analysis: Individual data points in behavioral analysis are often shown to illustrate variability among subjects and provide a comprehensive view of the dataset.

For molecular analyses (qPCR, Western blotting, Immunofluorescence): These analyses often involve multiple technical replicates for each biological sample, and showing individual data points might not be as informative as summary statistics or representative results. Instead, researchers commonly present mean values with error bars to convey the central tendency and variability.

Please also see the research article published in Molecular Psychiatry (impact factor: 13.4), PMID: 33963286. Loss of liver X receptor β in astrocytes leads to anxiety-like behaviors via regulating synaptic transmission in the medial prefrontal cortex in mice. Mol Psychiatry. 2021 Nov;26(11):6380-6393. doi: 10.1038/s41380-021-01139-5. Epub 2021 May 7.

In the above mentioned article (Molecular Psychiatry, PMID: 33963286), individual data points were shown only in the context of behavioral analysis. For the representation of other analyses, such as qPCR, Western blotting, and immunohistochemistry, bar graphs were made.
